# Tree species differ in plant economic spectrum traits in the tropical dry forest of Mexico

Marco V. Alvarado[1,2☯], Teresa Terrazas[1☯]*

1 Departamento de Botánica, Instituto de Biología, Universidad Nacional Autónoma de México, Ciudad Universitaria, Mexico City, Mexico, 2 Posgrado en Ciencias Biológicas, Universidad Nacional Autónoma de México, Ciudad Universitaria, Mexico City, Mexico

☯ These authors contributed equally to this work.
* tterrazas@ib.unam.mx

**Data Availability Statement:** Wood anatomical traits TDF raw data (DOI): 10.6084/m9.figshare. 24001932 Wood anatomical traits TDF means (DOI): 10.6084/m9.figshare.24001938.

## Abstract

In tropical dry forests, studies on wood anatomical traits have concentrated mainly on variations in vessel diameter and frequency. Recent research suggests that parenchyma and fibers also play an important role in water conduction and in xylem hydraulic safety. However, these relationships are not fully understood, and wood trait variation among different functional profiles as well as their variation under different water availability scenarios have been little studied. In this work, we aim to (1) characterize a set of wood anatomical traits among six selected tree species that represent the economic spectrum of tropical dry forests, (2) assess the variation in these traits under three different rainfall regimes, and (3) determine the relationships between wood anatomical traits and possible functional trade-offs. Differences among species and sites in wood traits were explored. Linear mixed models were fitted, and model comparison was performed. Most variation occurred among species along the economic spectrum. Obligate deciduous, low wood density species were characterized by wood with wide vessels and low frequency, suggesting high water transport capacity but sensitivity to drought. Moreover, high cell fractions of carbon and water storage were also found in these tree species related to the occurrence of abundant parenchyma or septate fibers. Contrary to what most studies show, *Cochlospermum vitifolium*, a succulent tree species, presented the greatest variation in wood traits. Facultative deciduous, high wood density species were characterized by a sturdy vascular system that may favor resistance to cavitation and low reserve storage. Contrary to our expectations, variation among the rainfall regimes was generally low in all species and was mostly related to vessel traits, while fiber and parenchyma traits presented little variation among species. Strong functional associations between wood anatomical traits and functional trade-offs were found for the six tree species studied along the economic spectrum of tropical dry forests.

**Funding:** TT funded by PAPIIT-UNAM in212622, MVA funded by CONACYT 810216 PAPIIT-UNAM, Programa de Apoyo a Proyectos de Investigación e Innovación Tecnológica, Universidad Nacional Autónoma de México. CONACYT, Consejo Nacional de Ciencia y Tecnología, https://dgapa.unam.mx/index.php/impulso-a-la-investigacion/papiit https://conacyt.mx/ The funders had no role in study design, data collection and analysis, decision to publish, or preparation of the manuscript.

**Competing interests:** The authors have declared that no competing interests exist.

## Introduction

Research on the functional diversity in ecological systems seeks to explain how species are assembled and perform in the environmental matrix (Reich, 2014 [1]). Each species possesses a set of traits that modulate its life history and performance within a specific ecosystem that acts as an environmental filter [2, 3]. It is thus vital to recognize the different types or functional profiles to understand the structure, function, and future of ecosystems [4].

In tropical dry forests, the functional or economic spectrum has been characterized based on how species use water, since water is the main limiting resource of these ecosystems [1, 5, 6]. This spectrum is defined by two functional extremes—species that avoid drought and species that tolerate drought [7, 8]. However, where a given species falls on this continuum is the result of the combination of several traits, resulting in a number of different functional classifications that condense the set of traits that contribute to the main functional profiles of this type of forest. These include classifications related to water use and species resistance to desiccation (isohydric/anisohydric species [7–9]); species that avoid drought and species that tolerate drought [8–11]; those related to growth rate, life history and wood density (colonizing species of rapid growth and low density of wood/successional species with slow growth and high wood density; [12–14]); and those associated with species' foliar phenological patterns, wood density and water storage [15–18].

All of these classifications confirm that species with robust xylem can generally tolerate strongly negative water potentials and are highly resistant to cavitation [16, 17]. Species at this end of the functional spectrum tend to have high wood density and be slow-growing, and since they have a resistant vascular system, they tend to be late-deciduous or facultative deciduous species. The species at the opposite end of the functional spectrum have xylem that is vulnerable to cavitation, but they compensate by regulating their water potentials to avoid the negative effects of drought on their sensitive xylem [2]. They are typically strictly drought-deciduous and tend to respond quickly to favorable environmental conditions by having low construction costs and therefore fast growth rates, which are associated with low wood density.

In the wood of angiosperms, water is transported through vessels in the xylem. Vessel diameter is the main trait that determines the water transport capacity of a given tree. As established by the Hagen–Poiseuille law, a one-unit increase in vessel diameter results in a fourfold increase in water conduction, so wider vessels have the capacity to carry a larger volume of water [19]. Furthermore, tree species with narrow vessels are often associated with lower hydraulic efficiency, but they tend to be more resistant to cavitation [5, 19, 20]. Although the trade-off between xylem safety and efficiency has long been recognized in plant hydraulics, recent research indicates that this trade-off is weak and that plants maintain both efficiency and safety within a certain threshold [21, 22]. However, data from temperate and boreal regions predominate in global analyzes while species from tropical dry forests are underrepresented [22–24]. Furthermore, in some of these analyzes data from branches were used and some authors mention that there is less variation in vessel traits coming from branches [21, 23], so these results cannot be generalized to all species and all types of forests.

Recent studies have consistently established that vessel diameter is not the only trait that determines variation in water conduction; other anatomical traits are also involved, but their relationships are not yet well understood [24]. Some studies suggest that increasing vessel diameter is mainly driven by an allometric relationship with plant height [25–27]. Other recent research suggests that the diameter and wall thickness of intervascular pits are important in the formation and propagation of air bubbles and that these are more important than vessel diameter [22, 28–30]. Still others show that xylem is a complex tissue in which mechanical

support, storage and water conduction functions are carried out by different cell types that are integrated into a limited morpho-functional space [23, 31]. Since the space that can be assigned to each cell type is limited, the cell fraction in xylem that is allocated to each cell type has important implications for hydraulic efficiency and safety, as well as the rise of functional trade-offs [32–37]. Tropical dry forest species are an ideal model in which to study this trait-based variation in the xylem. Since this type of forest experiences substantial annual water stress, species express multiple different xylem anatomical arrangements and a wide range of functional profiles that are reflected in the anatomical framework of their wood [35, 36, 38–40].

In this study, we aim to (1) quantify and assess the anatomical trait setup of the three xylem cell types across the economic spectrum of tropical dry forest species, (2) assess the variation in these traits under three different rainfall regimes, and (3) examine the relationships between anatomical traits and possible functional trade-offs. Our study includes six tree species that cover the full economic spectrum and three sites in the Mexican tropical dry forest with contrasting rainfall regimes. Five of the six selected species are distributed in all three study sites, so this work is one of the few that not only includes species from the economic spectrum but also allows the comparison of anatomical variation across different environments, an aspect that has been little explored, and provides data from tropical dry forests, forests that are often underrepresented in global analyses.

## Materials and methods

### Location and climatic elements of the study sites

We selected three study sites where tropical dry forest is the dominant vegetation. The sites fall along a latitudinal gradient over which total rainfall increases from north to south. The northernmost location was in Sierra de Manantlán in the state of Jalisco; the site Sierra de Montenegro was in Morelos; and the southernmost site was in Parque Nacional Huatulco in Oaxaca (S1 Table, Fig 1).

Red dots in the map indicate the location of each site. Walter-Liet diagrams were made with data from the nearest meteorological stations at each site (20333-Huatulco for Parque Nacional Huatulco; 17014-Temixco for Sierra de Montenegro; 14019-Autlán for Sierra de Manantlán). In the diagrams, the red dotted area indicates drought months in which the temperature exceeds precipitation. Vertical blue lines indicate water replacement, and the solid blue area indicates months in which the precipitation exceeds evaporation. The data were downloaded through the CLICOM system of the Centro de Investigación Científica y de Educación Superior de Ensenada, Baja California (CICESE). The map was made with ArcMap 10.5 [41].

The three sites are characterized by marked seasonality in rainfall (Fig 1). At all sites, the rainy season lasts five months (from June to October), and the dry season lasts from November to May. The dry season is characterized by high temperatures, especially from March to May, and monthly precipitation less than 100 mm. More than 80% of the annual precipitation falls during the rainy season. In addition to marked seasonality within years, there is high interannual variation in precipitation at all three sites. The highest mean values of total annual precipitation are found in Parque Nacional Huatulco (1,389.03 mm), and precipitation is usually above 900 mm, even during the driest years. The Sierra de Manantlán has the lowest average annual precipitation of the three sites at 677.26 mm, and during dry years, precipitation can be below 500 mm. Temperature varies little at the three sites (S1 Table). Huatulco has the highest average annual temperature and the average temperature of the hottest month (42.9°C in April, S1 Table). In contrast, Sierra de Manantlán has the lowest average annual temperature of the three sites and the lowest average temperature of the hottest month (S1 Table).

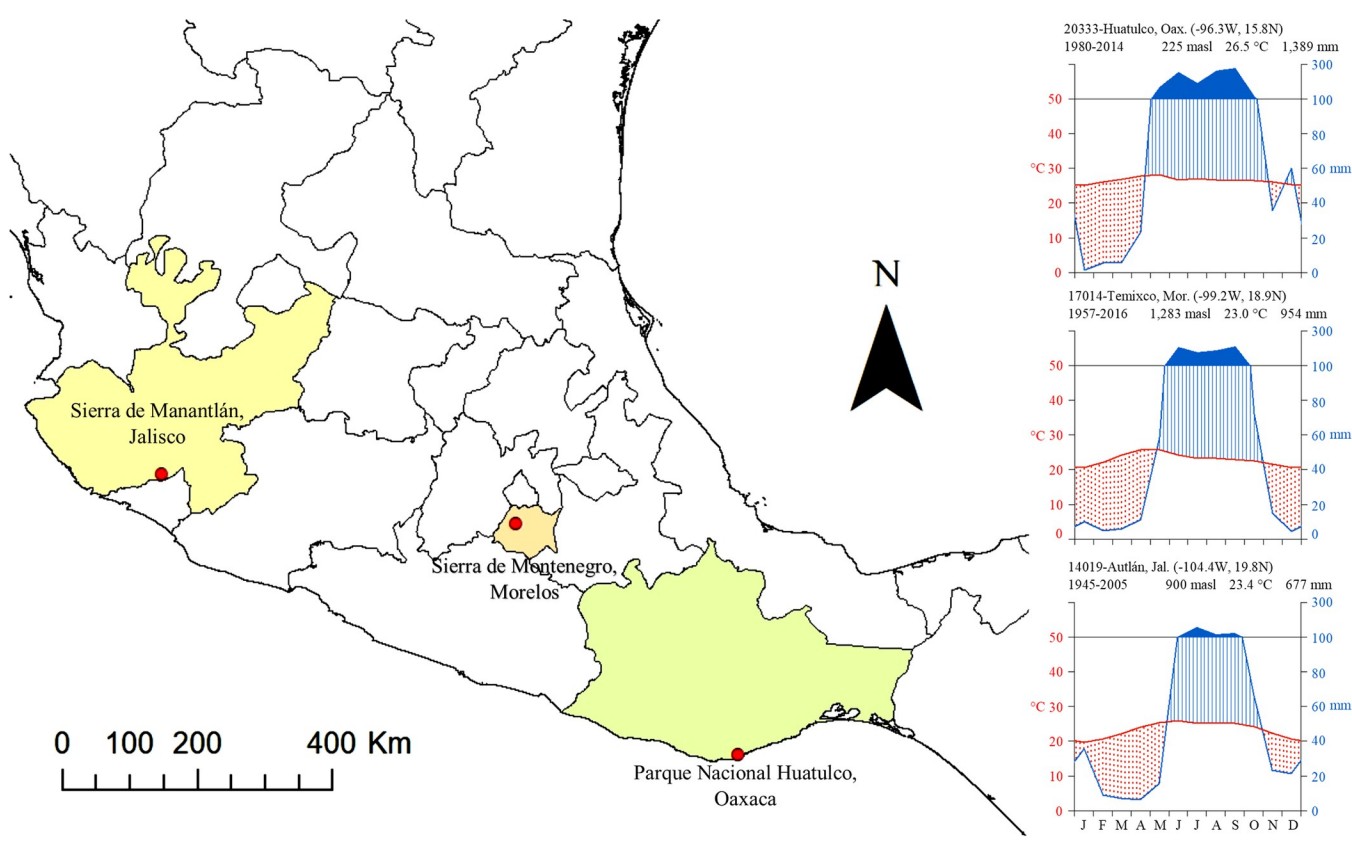

**Fig 1. Location of study sites and Walter-Liet climate diagram.**

## Selection of tree species, collection of plant material and processing of samples

We selected six tree species distributed in the three study sites. The selection criteria for the focal species were based on two proxies that have been used to characterize resource acquisition profiles: phenology and wood density (Table 1). We selected two drought-deciduous, low wood density tree species that represent drought-avoidance species; two hardwood tree species briefly to facultatively deciduous that represent drought-tolerant species and two medium

**Table 1. Selected tree species for the study and functional classification according to phenology and wood density.**

| .Species | Wood density (g cm⁻³) | Foliar phenology | Tree height (m) | References |
|---|---|---|---|---|
| **Succulent to drought-deciduous trees, wood density between 0.2–0.5 g cm⁻³ (group I)** | | | | |
| *Cochlospermum vitifolium* (Willd.) Spreng (Bixaceae) | 0.22 | strictly drought deciduous (succulent) | 8.4 ±3.6 | [18, 41, 44] |
| *Spondias purpurea* L. (Anacardicaeae)| | 0.35 | strictly drought deciduous | 7.4 ±1.5 | [18, 41–43] |
| **Briefly to late-deciduous trees, wood density between 0.4–0.7 g cm⁻³ (group II)** | | | | |
| *Pithecellobium dulce* (Roxb.) Benth. (Fabaceae) | 0.77 | late deciduous/ facultative deciduous | 8.1 ±1.5 | [41, 43, 44] |
| *Tabebuia rosea* (Bertol.) DC. (Bignoniaceae) | 0.72 | drought deciduous/ late deciduous | 14.8 ±1.7 | [18, 41, 42] |
| **Briefly to facultatively deciduous trees, wood density between 0.7–1.0 g cm⁻³ (group III)** | | | | |
| *Lysiloma divaricatum* (Jacq.) J.F. Macbr. (Fabaceae) | 0.92 | briefly deciduous/ facultative deciduous | 8.4 ±1.9 | [45] |
| *Haematoxylum brasiletto* H.Karst. (Fabaceae) | 0.95 | briefly deciduous/ facultative deciduous | 14.9 ±3.7 | [41–43] |

References are for phenology and wood density. Data are presented as mean height ± SD of the trees studied.

wood density briefly to late-deciduous trees that represent intermediate resource acquisition profiles between drought-avoidance species and tolerant species. As a coordination of these two proxies, we name three distinct groups that move along the plant economic spectrum of tropical dry forest. (Table 1). Unfortunately, two species were missing from one of the study sites (*T. rosea* in Montenegro and *H. brasiletto* in Huatulco). A voucher specimen for each species was housed at the Herbario Nacional de México (MEXU).

Wood samples from five reproductively active adult trees were collected per species at each site. We avoided trees with visible signs of damage or deformation. We tried to ensure that all the individuals were in similar ranges of height. Additionally, we measured the height of each tree since some studies have shown an allometric relationship between tree height and vessel diameter [25–27]. We used a saw to remove a wood sample from the trunk at a height of 1.5 m. The wood samples were collected during the dry season of 2022. Each wood sample was divided into two parts: one sample was used to calculate wood density, and the other was used for anatomical analysis. Wood density was estimated by the displacement method [46]. The weight of displaced water was obtained with a precision balance (Adventurer N13123, Ohaus, China, 0.001 g precision) and converted to volume. Samples were oven-dried at 105°C for 72 h to a constant weight. Wood density was calculated as the dry weight/fresh volume ratio. The wood sections for the anatomical analysis were immediately fixed in formalin–acetic acid–alcohol solution [47]. In the laboratory, sections (transverse, tangential and radial) 18–20 $\mu$m thick were cut with a sliding microtome (Leica, SM2000R, Germany). The sections were dehydrated in a series of alcohols from 50, 70, 96 to 100%, double stained with safranin and fast green and mounted on slides with synthetic resin [47]. For observation and measurement, we used an Olympus B51 microscope with an Infinitum camera.

## Determination of wood anatomical traits and statistical analyses

Table 2 lists and describes the wood traits measured for the 80 wood samples. We selected 12 traits to understand the relationship between different wood cell fractions and their relationships with vessel traits, fiber size and wall thickness and wood density between the six tree species of the tropical dry forest. Vessel diameter and frequency are the main descriptors in wood anatomy of indirect water conduction capacity and safety as a redundancy factor [48, 49]. The

**Table 2. Wood traits measured in six tree species from the tropical dry forest.**

| Trait | Description |
|---|---|
| Vessel radial diameter ($V_D$) | Mean vessel radial diameter ($\mu$m) |
| Vessel frequency ($V^{mm-2}$) | Number of vessels per unit area (1 mm$^{-2}$) |
| Vulnerability Index (VI) | A robust indicator of vulnerability to drought-induced cavitation. VI = $V_D$ / $V^{mm-2}$ |
| Vessel fraction ($V_F$) | Fraction of vessels in cross-section |
| Fiber diameter ($F_D$) | Mean fiber diameter ($\mu$m) |
| Fiber lumen diameter ($F_{Dl}$) | Mean fiber lumen diameter ($\mu$m) |
| Fiber wall thickness ($F_{WT}$) | Mean fiber wall thickness ($\mu$m) |
| Fiber fraction ($F_F$) | Fraction of fibers in cross-section |
| Axial parenchyma fraction ($P_{Fa}$) | Fraction of axial parenchyma in cross-section |
| Radial parenchyma fraction ($P_{Fr}$) | Fraction of radial parenchyma in cross-section |
| Total parenchyma fraction ($P_F$) | The total fraction of both, axial and radial parenchyma in cross section. $P_p = P_{pa} + P_{pr}$ |
| Wood density (WD) | Mass of wood for unit of area (g cm$^{-3}$) |

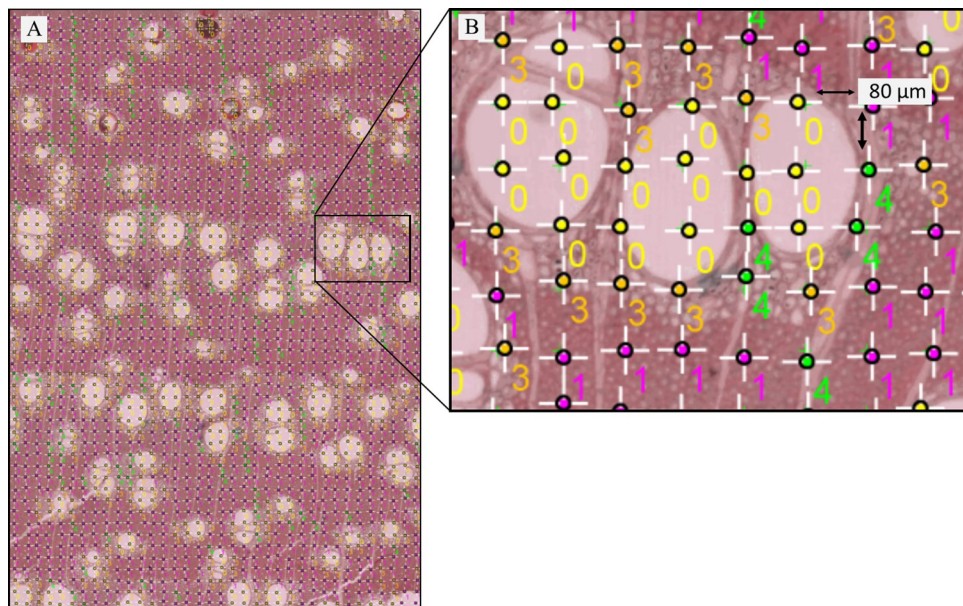

**Fig 2. Image analysis of wood cross-sections for cell fraction measurements.** A Full image showing the grid and the marked points for each cell type (11.76 mm$^2$, 1872 grid points). B. Close-up showing the arrangement of the points and distance between each grid point. In this example, 0, vessels; 1, fibers; 3 axial parenchyma; 4, radial parenchyma.

vulnerability index [50] estimates the degree of xeromorphy of the xylem and is the anatomical counterpart of water potential at 50% conductivity loss [51]. Fiber cell diameter, lumen diameter, and wall thickness are associated with xylem resistance to negative pressures [32, 33, 52]. For vessel diameter, fiber cell diameter, lumen diameter, and wall thickness, we took 50 measurements per sample along the radial axis of the wood from multiple growth rings. We start at one end of the sample, in the outermost part of the wood close to the vascular cambium. From there, we advanced in a radial direction. Once the limit of the sample was reached, we returned to the outermost part to continue scanning the section until reaching 50 vessels. Vessel frequency was calculated by counting all vessels in ten 2.2 mm$^2$ cross-sections. Similar to vessel diameter, we started at one end of the sample, in the outermost part of the wood close to the vascular cambium and moving radially. It was ensured that each measurement did not include the area of the previous section. Once all the vessels have been counted, the number of vessels per mm$^2$ is expressed [48, 49]. All measurements were performed with an IMAGE-Pro Plus (Media Cybernetics) image analyzer ver. 7.1.

To determine the fraction of each cell type in the cross-section, we followed the method of Ziemińska and collaborators [52]. To do this, the area of analysis ranged from 5 mm$^2$ to 12 mm$^2$, depending on the amount of tissue available for each wood sample. The distance between grid points was 80 $^2$m, and we analyzed between 900 and 1,800 points depending on the amount of tissue available (Fig 2). ImageJ version 1.5 was used to calculate cell fractions [53]. Each cell type (vessels, fibers, axial parenchyma and radial parenchyma) was identified by a color and number mark. If a grid point was on the lumen or wall of a cell type, it was counted as such. The sum of each cell type divided by the total points was the proportion of each cell type.

To test differences in wood anatomical traits among species and sites, we fitted linear mixed models (LMM) based on raw rata for each of the following response variables: vessel diameter, vessel frequency, fiber cell diameter, lumen diameter, and wall thickness. A full model included the interaction of tree species (categorical, six levels) and sites (categorical,

three levels) as fixed effects and individuals as the random intercept effect. In the subsequent models, the interaction was eliminated, the effect of each factor was tested separately (one model with only the effect of the species and another with only the effect of the site), and a null model was also fitted in which species and site were not included. Subsequently, model selection was made using two approaches. Akaike's information criterion (AIC) and the likelihood ratio test (LRT). For models in which LRT showed statistically significant goodness of fit and when ΔAIC<10, estimated marginal means (EEM) were calculated, and multiple pairwise comparisons were performed to contrast the variation between species and sites. For the vulnerability index, wood density and cell fractions, a two-way ANOVA was performed to evaluate differences between sites and species from functional groups. Post hoc comparisons were performed using Tukey's HSD test when significant differences were detected. We applied a Shapiro–Wilk test to test for normality distribution and Levene's test for homoscedasticity. Since the raw variables did not fulfill the assumptions of normality and homoscedasticity and to reduce the effect of outliers, we used log-transformed response variables for the statistical analyses.

To explore trade-offs between wood traits, we calculated Pearson's correlations for all traits with full data of the species studied. Additionally, to quantify the overall variation in each species, we estimated the relative distance plasticity index (RDPI) for anatomical traits [54]. RDPI is used to calculate the distance in one trait between two individuals of a single species under different conditions. RDPI tests the overall plasticity of a species. In our study, the three sites represent different conditions. RDPI measures plasticity in wood anatomical traits of a single species across the three different rainfall regimes. All analyses were performed in R version 4.1.2 [55]. To construct the LMMs, we used the package lme4 [56], and to estimate the Valladares index, we used the 'Plasticity' package [57].

## Results

S2 Table summarizes the values obtained for each anatomical trait for the six species studied at the three study sites. Fig 3 illustrates the wood anatomy of each species. For vessel and fiber traits, the mixed models highlight species and site interaction as the main driver of variation, since the best supported model for both the AIC (the lowest value of all models) and with the highest level of statistical significance for LTR ($p<0.001$, S3–S5 Tables). The second model with the best support for both approaches was the model that only includes the effect of species. The null model and the model with only site as a factor had the highest AIC values and were not statistically significant as a result of LRT (S3 Table). The model in which the effect of both factors was added was not supported by LTR and was not statistically significant. The penalty by LTR can be explained because the effect that the site adds by itself explains little variation, so for the TLR, adding this effect without interaction penalizes the model and is not significant.

For the vulnerability index, wood density, and the three cell fractions, statistically significant differences were found for species and the interaction between species and site but not for site (S6 Table). Since both the mixed models and the ANOVA showed that the effect of the species by itself has an important effect on the variation in wood anatomical traits, the results by species will be presented first, followed by the interaction between species and site.

### Differences in wood anatomical traits along the plant economic spectrum of tropical dry forest

For the mixed models, according to the multiple pairwise comparisons in the EMM (S5 Table), *Cochlospermum vitifolium*, group I (Fig 4A), had the widest vessels, with a mean vessel diameter of 237.5±47.0 $^2$m, which differed from that of the other species. Species of group III

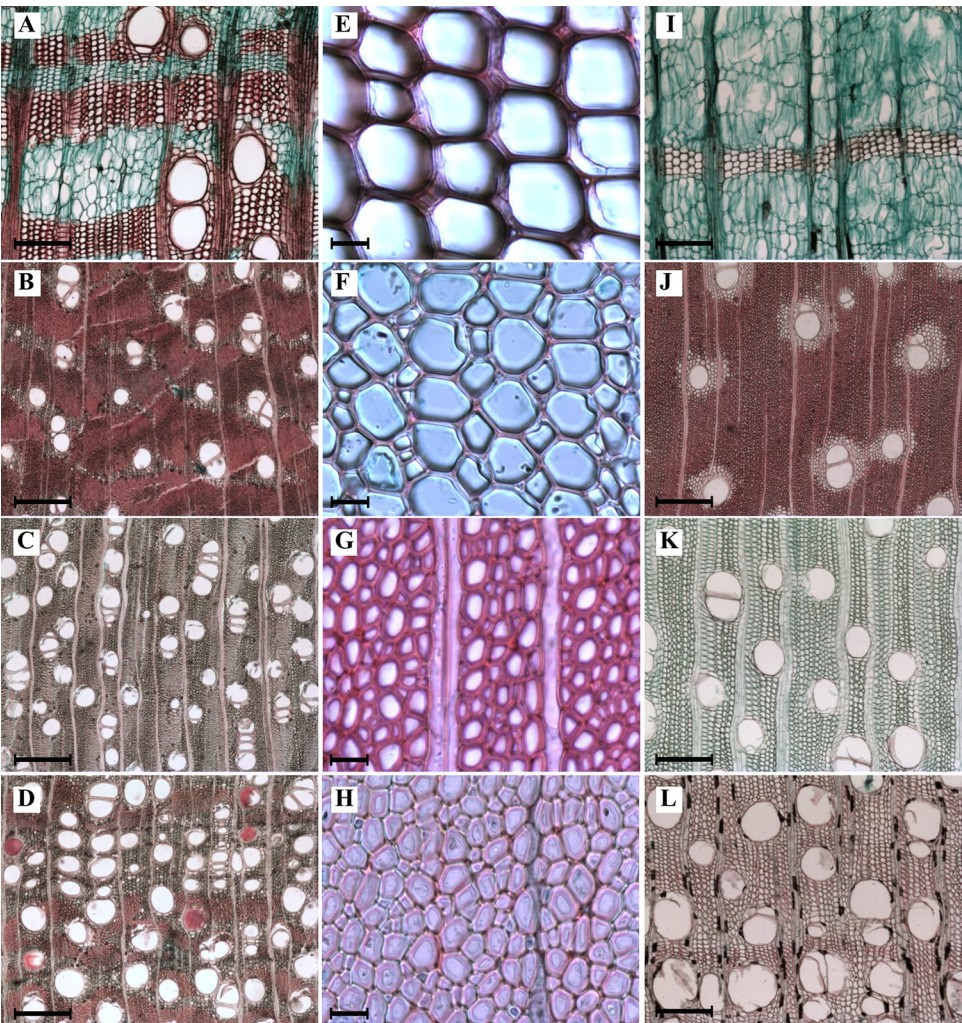

**Fig 3. Variation in wood anatomy among the studied tree species. A.** *Cochlospermum vitifolium* had the largest vessels and the lowest density. **B**. *Lysiloma divaricatum* had narrow vessels and the largest proportion of fibers. **C-D.** The highest vessel density was associated with denser wood species, group III (**C**, *L. divaricatum*; **D**, *Haematoxylum brasiletto*). **E-F.** The largest diameter and thin-walled fibers were found in strict drought deciduous species with low wood density, group I (**E**, *C. vitifolium*; **F**, *Spondias purpurea*). **G-H.** Narrower, thick-walled fibers were associated with high wood density species, groups II and III (**G**, *H. brasiletto*; **H**, *Pithecellibium dulce*). **I.** *C. vitifolium* showed nonlignified parenchyma and the highest fraction of this cell type. **J.** *L. divaricatum* showed the lowest parenchyma fraction and the highest fiber fraction. **K**. *S. purpurea* had a high proportion of thin-walled fibers. **L**. *S. purpurea* individuals with a vessel fraction above 20%. Bar is 300 μm in A-D, I-L; 20 μm in E-H.

had the lowest mean vessel diameter of the three groups, being the narrowest in *Lysiloma divaricatum* with 135.8±36.5 $^2$m (Fig 4A). Species of group II had intermediate values between groups I and III. The highest values of vessel frequency were found in *L. divarivatum* and *H. brasiletto* (group III).

Differences between species were also found for fiber traits. The highest mean values for fiber diameter and lumen were found in species belonging to group I (Fig 4C). Fiber wall thickness differed among species. Thin-walled fibers were found in species of group I, which were significantly different from those in species of groups II and III (Fig 4D).

Wood cell fractions differed among species (Fig 5A and 5B; S6 Table). For the fiber fraction, *C. vitifolium* had the lowest value and was significantly different from the rest of the species

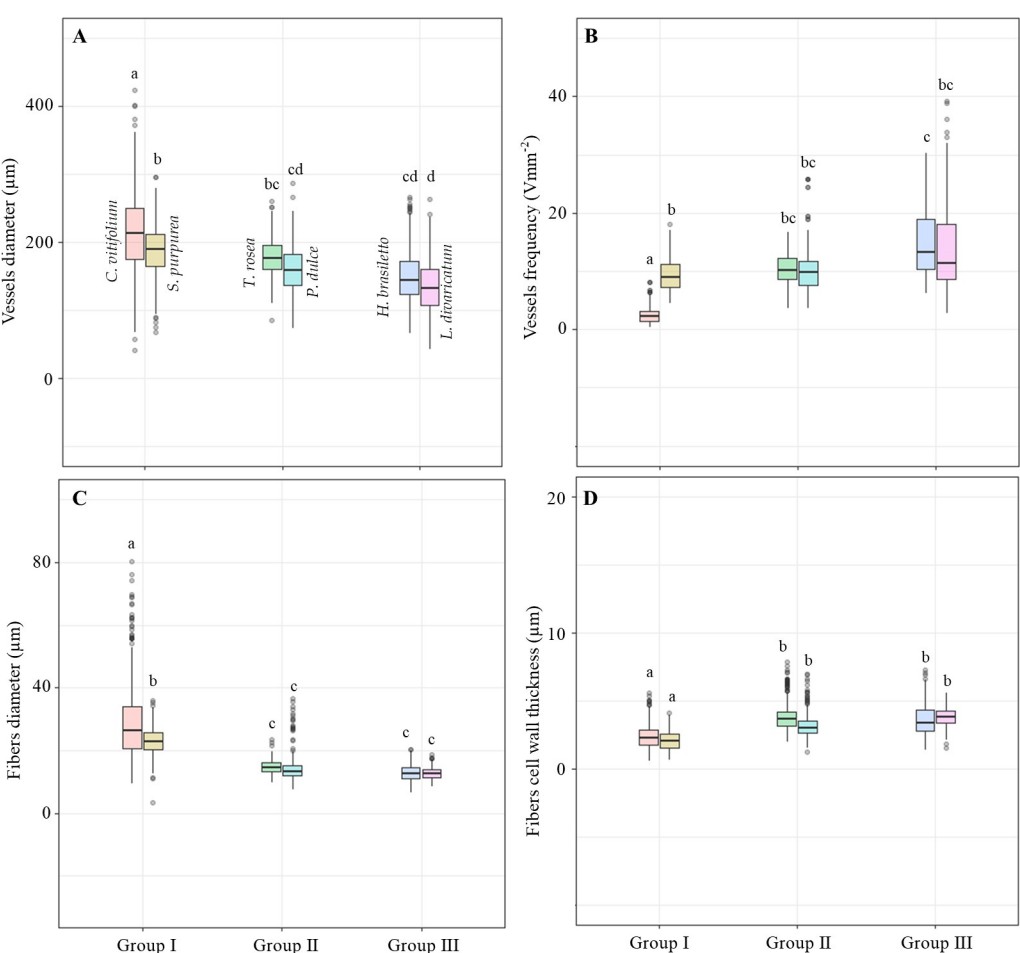

**Fig 4. Differences in wood anatomical traits among species.** Boxplots representing full data of wood anatomical traits for each tree species studied. Different letters indicate significant differences among species, according to multiple pairwise comparisons resulting from mixed models. A. Vessel diameter. B. Vessel frequency. C. Fiber cell diameter. D. Fiber wall thickness. Details of the mixed models can be found in S3 and S5 Tables.

(Fig 5B). *L. divaricatum* (group III) and *S. purpurea* (group I) did not show statistically significant differences (Fig 5B). For the parenchyma fraction, the highest value was found in *C. vitifolium* and was significantly different from the rest of the species (Fig 5B). The vessel fraction presented statistically significant differences only for *C. vitifolium* (group I) with respect to the remaining species, which had the lowest vessel fraction (Fig 5B). Vessels represented the smallest fraction of the morpho-functional space in species wood, rarely exceeding 30%.

Vulnerability and wood density presented statistically significant differences. For the vulnerability index, differences were found only for *C. vitifolium* (group I), which had the highest vulnerability index (106.2 ±38.5). The lowest wood density was found in *C. vitifolium* and was significantly different from that of the other species (Fig 5C). The highest wood density was found in species of group III, which was significantly different from the other species (Fig 5C).

## Differences in wood anatomical traits among sites

For vessel diameter, statistically significant differences in EMM were detected among sites only for *C. vitifolium* and *H. brasiletto* (Fig 6A, S4 Table). For *C. vitifolium*, trees in Sierra de Montenegro (Montenegro henceforth) had narrower vessels than those in Parque Nacional

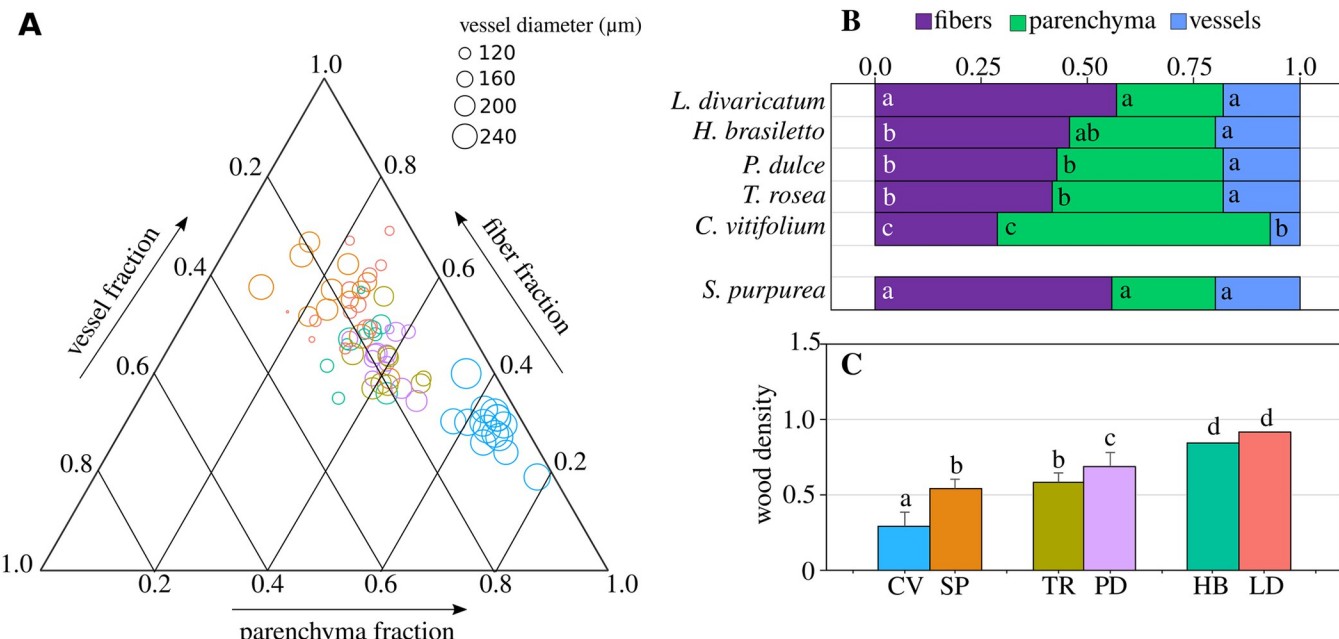

**Fig 5. Wood cell fraction and wood density variation among species.** A. Ternary plot showing the distribution of tree species according to cell fraction. Each circle represents mean values for each sampled tree, and circle sizes indicate mean vessel diameter. B. Cell fraction distribution in the six tree species from the tropical dry forest. *S. pupurea* is placed separately due to its high proportion of fibers, but these are thin-walled septate fibers that store starch grains. C. Mean wood density in each of the species. For cell fraction and wood density, different letters indicate significant differences among species. Colors in A are similar to those in C and indicate each tree species. Cv, *C. vitifolium;* Sp, *S. purpurea*; Tr, *T. rosea*; Pd, *P. dulce*; Hb, *H. brasiletto*; Ld, *L. divaricatum*.

Huatulco (Huatulco henceforth) and Sierra de Manantlán (Manantlán henceforth), but no differences were detected between those in Huatulco and Manantlán for this species. *H. brasiletto* had larger vessels in Montenegro than in Manantlán. For vessel frequency, differences in EMM were detected among sites only in *L. divaricatum* and *H. brasiletto* (Fig 6B). For *L. divaricatum* vessels, the frequency was higher in Huatulco than in Manantlán and Montenegro. For *H. brasiletto*, higher vessel frequencies were found in Manantlán than in Montenegro.

For fiber diameter differences in EMM were detected only among sites for *C. vitifolium* (Fig 6C). Montenegro had narrower fibers compared to Huatulco and Manantlán. No differences were found among sites for any species for fiber lumen diameter. Differences among sites for fiber cell wall thickness were detected only in *H. brasiletto* (Fig 6D). Trees from Manantlán had thicker fiber cell walls than those from Montenegro.

The vulnerability index showed statistically significant differences among sites only for *C. vitifolium* (Fig 7A, S4 Table). Higher vulnerability was found in Huatulco than in Montengro and Manantlán. For wood density, we found differences among sites only for *H. brasiletto*, and denser wood was found in Manantlán than in Montenegro (Fig 7B). For vessel fraction, statistically significant differences were found only for *L. divaricatum* (Fig 7C); trees from Huatulco had a higher vessel fraction than those from Montenegro and Manantlán. For the fiber fraction and parenchyma fraction, no statistically significant differences were detected among sites for any species (Fig 7D and 7E).

## Correlation between wood anatomical traits

We found strong correlations between traits in a way that reflects the trade-offs that exist between the functions in wood and the differences between functional groups previously

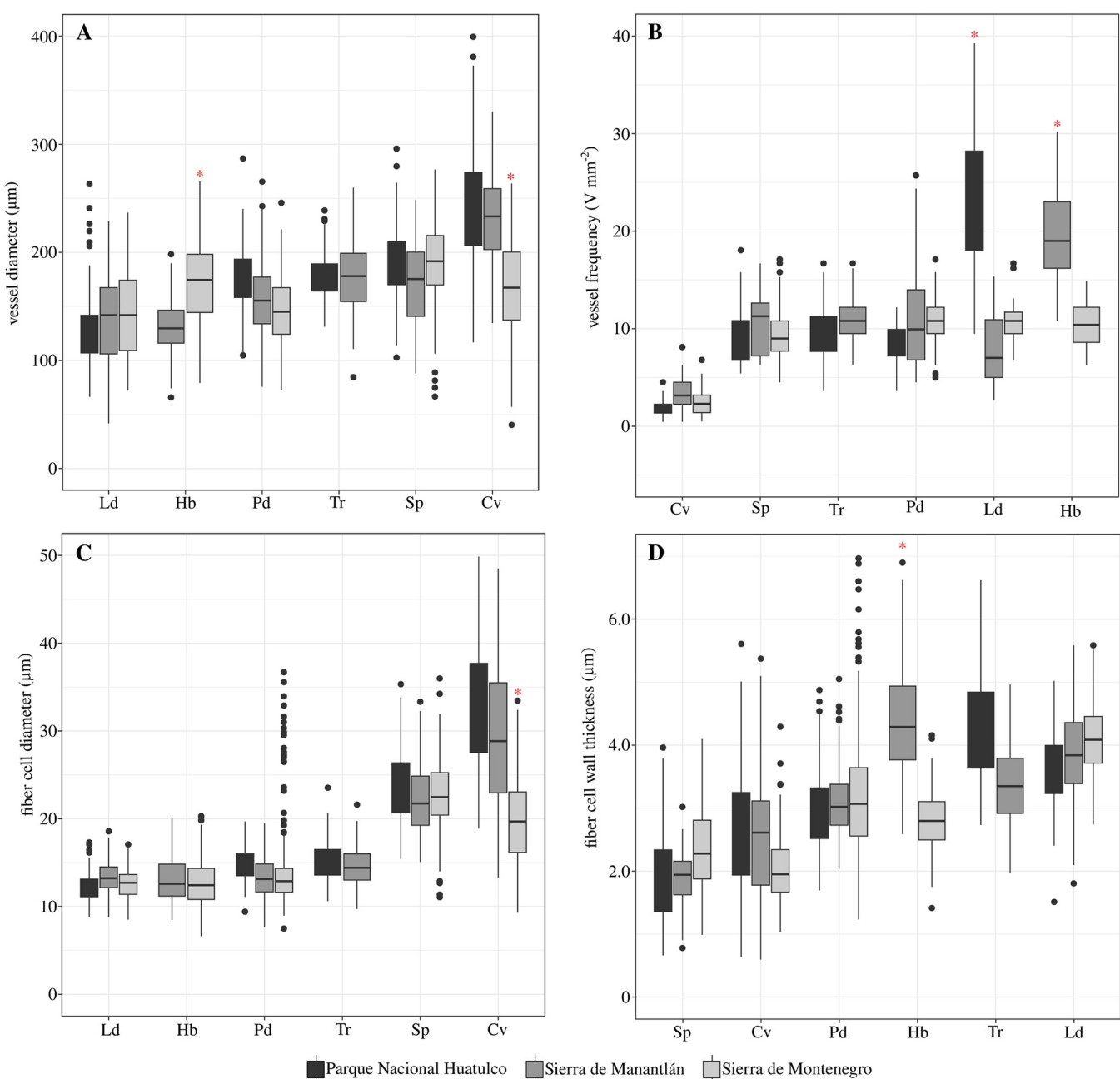

**Fig 6. Variation in wood anatomical traits among sites from multiple measurements.** Boxplots showing the values obtained for the wood anatomical traits in each species across the three study sites. The color in the boxplot represents each site. Ld, *L. divaricatum*; Hb, *H. brasiletto*, Pd, *P. dulce*; Tr, *T. rosea*; Sp, *S. purpurea*; Cv, *C. vitifolium*. Asterisks indicate the level of significance of the differences between sites as a result of the multiple pairwise comparisons in the EMM. *, <0.001. For the rest of the comparisons, no significant differences were found. Details of the mixed models can be found in S3 and S5 Tables.

mentioned. S7 Table shows the correlation matrix between all pairs of traits studied. Most of the variables had correlation coefficients greater than 0.5 and were statistically significant (Fig 8), and a negative and positive correlation was found for vessel diameter and total parenchyma fraction. With full data, a weak correlation was found, and when performing the correlation analysis without *S. purpurea*, the correlation increased to r = 0.87 (Fig 8C). For the total parenchyma fraction, the correlation analysis excluding *S. purpurea* showed a stronger correlation

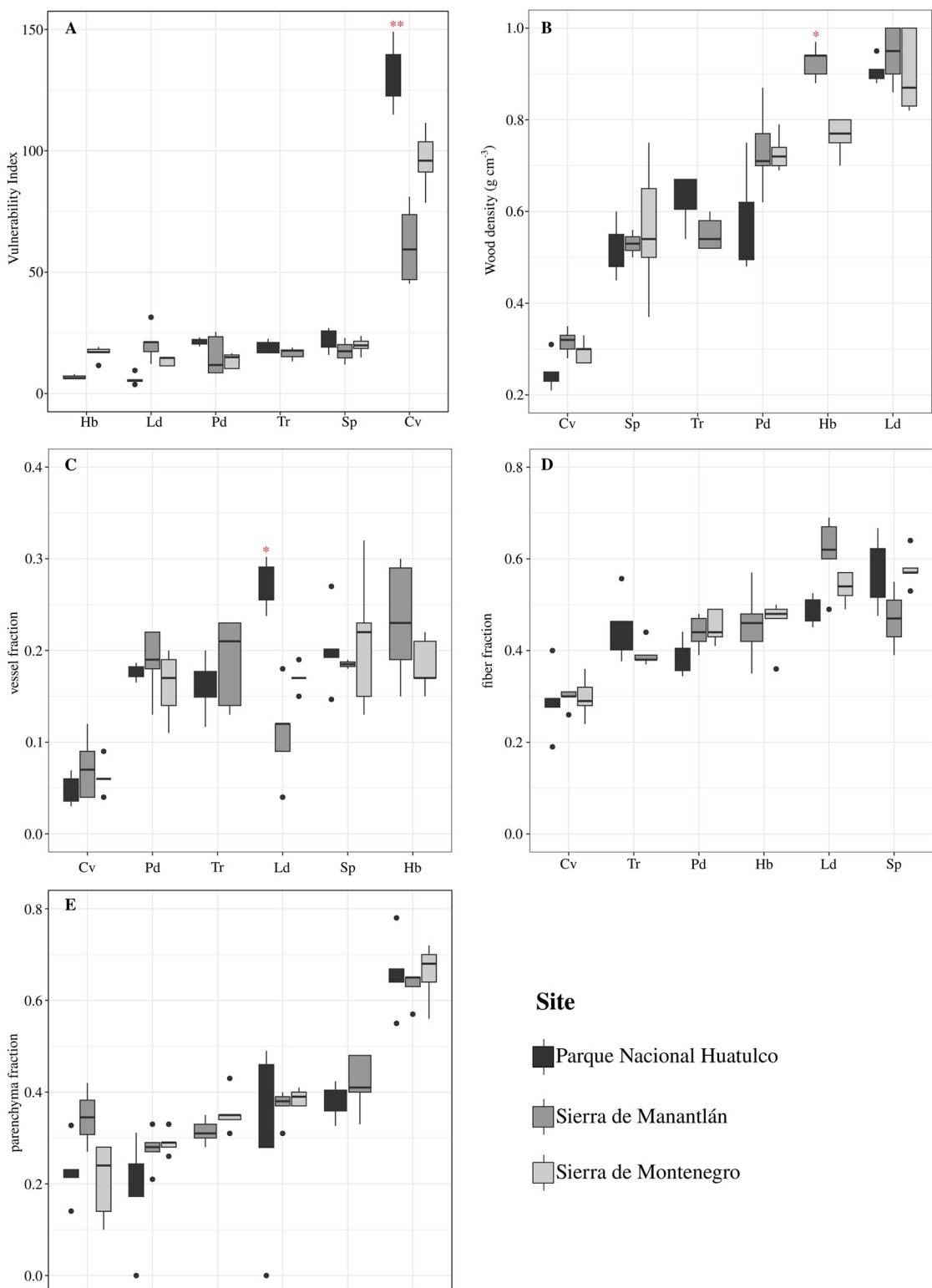

**Fig 7. Variation among sites in wood traits from a single measurement.** Boxplots showing values obtained from A, vulnerability index; B, wood density and C-D, the three cell fractions. The color in the boxplot represents each site. Ld, *L. divaricatum*; Hb, *H. brasiletto*, Pd, *P. dulce*; Tr, *T. rosea*; Sp, *S. purpurea*; Cv, *C. vitifolium*. Asterisks indicate the level of significance of the differences between sites. **, <0.01; *, <0.05.

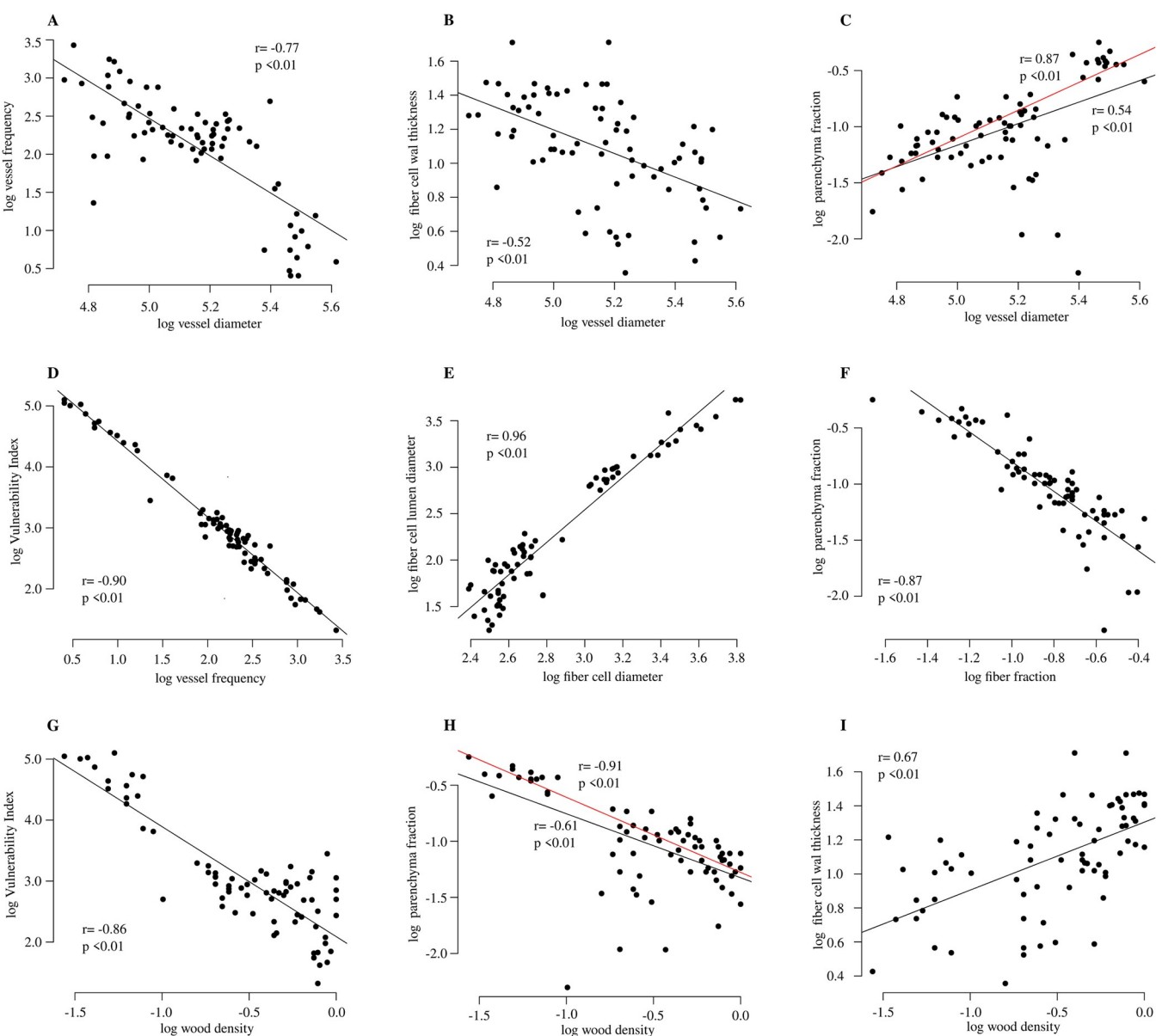

**Fig 8. Correlations between wood anatomical traits.** Each point represents the mean value of a single tree. Fitted linear regression. r, Pearson's correlation. In C and H, the red line and associated values were estimated by removing data from *S. purpurea* (see text for details).

with wood density (r = -0.91) than the analysis including *S. purpurea*. Tree height showed no statistically significant correlation with vessel diameter.

## Phenotypic plasticity in wood anatomical traits

Wood anatomical traits showed low phenotypic plasticity (RDPI<0.4) for all species (Table 3). *C. vitifolium* and group III species had the highest plasticity values. In *C. vitifolium*, the highest plasticity was recorded for vessel frequency (0.316), vulnerability index (0.252) and fiber cell and lumen diameter (0.238; 0.272). For *H. brasiletto* and *L. divaricatum*, vessel frequency (0.294 and 0.352, respectively) and vulnerability index (0.412 and 0.385, respectively) had the highest plasticity values. Additionally, for fiber lumen diameter, *H. brasiletto* showed the

**Table 3. Relative distance plasticity index (RDPI) for wood anatomical traits in each species studied.**

| Trait | *C. vitifolium* | *S. purpurea* | *T. rosea* | *P. dulce* | *H. brasiletto* | *L. divaricatum* |
|---|---|---|---|---|---|---|
| $V_D$ | 0.170 | 0.119 | 0.083 | 0.129 | 0.162 | 0.155 |
| $V^{mm-2}$ | 0.316 | 0.164 | 0.147 | 0.195 | 0.294 | 0.352 |
| VI | 0.252 | 0.138 | 0.095 | 0.236 | 0.412 | 0.385 |
| $V_F$ | 0.214 | 0.137 | 0.147 | 0.108 | 0.151 | 0.304 |
| $F_D$ | 0.238 | 0.103 | 0.078 | 0.109 | 0.116 | 0.075 |
| $F_{DI}$ | 0.272 | 0.132 | 0.178 | 0.179 | 0.273 | 0.167 |
| $F_F$ | 0.088 | 0.084 | 0.072 | 0.066 | 0.079 | 0.088 |
| $P_F$ | 0.051 | 0.218 | 0.080 | 0.062 | 0.069 | 0.096 |
| WD | 0.094 | 0.096 | 0.072 | 0.100 | 0.096 | 0.04 |

highest RDPI value (0.273), and *L. divaricatum* showed the highest RDPI value for vessel fraction (0.304). Three species—*S. purpurea*, *T. rosea* and *P. dulce*—had the lowest phenotypic plasticity in all of their wood anatomical traits (<0.2).

## Discussion

### Wood anatomical traits characterize plant economic spectrum in tropical dry forest

The strictly drought deciduous species (group I) showed an important differentiation with respect to the rest of the species. *C. vitifolium* and *S. purpurea* presented a rapid resource acquisition profile similar to other strict deciduous species [1, 5, 17, 58]. These species have the widest vessels (especially *C. vitifolium*, with vessel widths up to 400 $^2$m) and the lowest frequencies. However, vessel frequency in *S. purpurea* was not significantly different from the rest of the species in the other two groups. This combination of traits characterizes hydraulic systems with a high capacity for water conduction through few vessels with wide lumina [19, 59–61], which has been recorded for other strictly drought deciduous species [34, 62, 63]. However, the cavitation risk of these few vessels is high [59, 60], as evidenced by the vulnerability index values in this group. High values of this index are indicators of high sensitivity to water stress [17, 51, 63, 64] and are a characteristic of strictly drought deciduous trees in tropical dry forests [64].

Despite this significant vulnerability, both *C. vitifolium* and *S. purpurea* are abundant and widely distributed species in neotropical forests [65, 66]. In our results, *C. vitifolium* assigned the largest fraction of wood volume to nonlignified parenchyma. The role of parenchyma in coping with water stress and seasonal changes in water has been extensively studied [38, 67]. A study of the stems of *C. vitifolium* trees [68] found that nonlignified parenchyma stores large amounts of water, nonstructural carbohydrates and mucilaginous substances that act as a water source, control pressure variation in vessels and maintain high water potentials. It is likely that the hydraulic system of this species has a risk of cavitation. However, it maintains water potentials due to the early loss of leaves after the end of the rainy season and support of the hydraulic system by nonlignified parenchyma [35, 40, 58, 59, 69]. In *S. purpurea*, the largest fraction of its wood volume is occupied by fibers, while axial parenchyma is almost absent. Traditionally, fibers have been associated with the function of mechanical support and resistance of the wood associated with its significant degree of lignification [35, 52, 59, 69]. However, thin-walled septate fibers have been related to water and nonstructural carbohydrate storage and mobilization [63, 69]. It is possible that thin-walled septate fibers in *S. purpurea* act not only as mechanical support but also fulfill storage functions, making them analogous to

the nonlignified parenchyma of *C. vitifolium*. In summary, in this functional group, the large amount of storage tissue, in conjunction with the rapid loss of leaves and an efficient hydraulic system, allows rapid resource acquisition during the rainy season and minimizes cavitation risk during the dry period. Similar observations were found in strictly drought deciduous trees of the Brazilian Caatinga and Cerrado [63, 69].

Along the economic spectrum, hardwood facultative deciduous tree species (group III) had the narrowest vessels, which tend to characterize species that are most resistant to cavitation, as many studies point out [19, 59, 60]. Water conduction is reduced, and the risk of the rupture of the water column decreases [59]. Vessel frequency was numerically highest in this group (statistically significantly higher than in group I species but not significantly different from group II). This combination of traits has already been characterized in other hardwood species [58, 63, 70, 71] in physiological studies that show greater tolerance to loss of conductivity and greater tolerance to more negative water potentials [6, 8, 17, 58, 72].

Moreover, in group III species, fibers had narrow diameters and thick walls and the highest values for fiber fractions in the species studied. Several studies have shown that highly desiccation tolerant and high wood density long-lived species have high fiber fractions with very thick walls [51, 73–76], similar to the species of group III. As several authors suggest, fibers not only serve as the mechanical support of wood but also function as a matrix that gives hydraulic resistance to the vessels and allows them to withstand more negative pressures [32, 33, 35, 61, 69, 71, 77]. In relation to axial parenchyma, species of group III had the smallest fractions of this cell type; however, parenchyma is in contact with vessels (vasicentric to confluent). Some authors suggest [36, 73–75, 78] that parenchyma cells in contact with vessels have larger pits, favoring hydraulic capacitance. Thus, although the parenchyma fraction is low in group III species, the vessels have external support during the most stressful periods. In future physiological studies, it is necessary to determine whether parenchyma acts as an active secondary support for vessels in species such as group III, despite its limited fraction.

It is interesting that briefly to late-deciduous trees, with medium wood density (0.4–0.7 g cm$^{-3}$) showed intermediate values between species of groups I and III. *T. rosea* does it towards group I sharing similarities with *S. pupurea* and *P. dulce* with both species of group III. This trend seems to support that the plant economic spectrum is a continuum between both extremes, from rapidly deciduous species with very sensitive hydraulic systems to drought-tolerant species [6, 78, 79]. In *P. dulce* and *T. rosea*, the distribution of the morpho-functional space to fibers and parenchyma presents a very similar distribution. This indicates **that** the functions of mechanical and hydraulic resistance as well as storage and vessel traits are well balanced in the intermediate species of the economic spectrum. In other studies, intermediate species show high plasticity in wood traits or traits that do not closely conform to either functional extreme (between obligated deciduous/succulent trees and facultative/evergreen species [63, 69, 80]).

## Low variation in wood anatomical traits among rainfall gradients in tropical dry forests

Contrary to our expectations, wood anatomical traits presented low variation and plasticity among sites. Although our sites represent different rainfall regimes, it is important to highlight that our study compares forests of the same type. Unlike other types of forests, in tropical dry forests, the asymmetric annual distribution of rain is the main environmental variable that controls the structure of the vegetation. Precipitation also acts as the principal selection pressure that shapes and determines variation in plant traits, including wood traits [40, 81–83]. It is likely that the high interannual variation in precipitation that occurs in these forests shapes

the xylem. For the species studied here, we found high intraspecies variation that could be the result of interannual variation, which possibly explains why low variation between sites was found. Similar results were found in other studies that compared rainfall differences in tropical dry forests [84, 85]. Instead, differences are found when different types of forests are compared [37, 85–87].

The main differences that we found are concentrated in vessel traits, and there is a slight tendency toward larger vessels and a lower frequency (hence greater vulnerability) at the site with the highest total annual rainfall (Huatulco) and narrower vessels and a greater frequency at the site with the lowest total annual precipitation (Manantlán). In other studies that compare rainfall differences in tropical dry forests, vessel traits show the main variation [84, 88].

Furthermore, RDPI also presented low values in all species for vessel diameter. The comparison of RDPI values between studies is still limited. For example, for a tropical dry forest study, similar RDPI values were found [85], notably low plasticity in vessel RDPI values between sites for *T. rosea*. In a more recent study with temperate oak species along an altitudinal and humidity gradient [89], similar vessel RDPI values were found in all *Quercus* species studied, although vessel diameter and frequency had the greatest variation in all their studied traits.

Contrary to our expectation, *C. vitifolium* from group I presented the greatest variation in all its wood anatomical traits. Our study reveals that succulent species present much more variation than what was previously expected and that succulent trees also exercise significant control over vessel traits [64, 69, 70]. Some authors propose that in drought-avoidant species, the adjustment of vessel traits such as diameter and frequency is important to address seasonal changes in water availability to protect the xylem from changes in water potential [61, 62, 69, 90].

The cell fraction presented low variation among sites for all species; statistically significant differences were found only in *L. divaricatum* for the vessel fraction, and their RDPI values were the lowest of all the traits studied. It is important to mention that statistically significant variation in fiber and parenchyma fractions has been found [37, 82, 86]. For example, in a study along an aridity gradient [82], in drought-avoidant species, the parenchyma fraction is higher at the driest sites, potentially offering greater control over the conduction network since a greater fraction of parenchyma offers support to the vessels against the scarcity of water and capacitance control.

Nevertheless, in perennial hardwood species, the fiber fraction is higher at drier sites since the vessel network needs a resistance matrix to face the high negative pressures that xylem experiences in arid places. In contrast, in temperate oak species, fiber and parenchyma fractions varied together according to changes in an altitudinal and precipitation gradient [37]. In most of the *Quercus* species studied, the parenchyma fraction was higher and the fiber fraction was lower the wetter the site. According to the authors, oak species invest in xylem safety as water stress increases during spring, when high temperature and low precipitation determine the highest level of water stress. These studies seem to suggest that the differences between sites do not present a unique trend in relation to parenchyma and fiber fractions; rather, they seem to be more specific regarding the functional profile of each species and the climatic variation of each environment. Since we did not find statistically significant differences in our data for fiber and parenchyma fractions between sites and there was no unique trend between functional groups, we cannot adequately explain the variation in these cell types in tropical dry forests. We hypothesize that since the mechanical-hydraulic resistance of wood and the storage capacity are highly compromised in tropical dry forests [34, 36, 51, 62, 73], there is low variation in these functions, and therefore, the greatest selection operates on the vessel traits.

## Wood anatomical traits explain functional wood trade-offs along the economic spectrum in tropical dry forests

Since we found mostly negative correlations, the results indicate that there is an important trade-off between functions in wood and that the morpho-functional space available to each cell type has important effects on the functional profile of a species.

The trade-off between water conduction and safety (i.e., diameter versus frequency of vessels) is one of the most important wood structural trade-offs [19, 60]. In our analysis, the distribution of the individuals of each functional group in relation to vessel diameter and frequency follows this trade-off between water conduction and hydraulic safety. Water conduction and safety also showed clear relationships with storage and mechanical resistance functions. Vessel diameter was positively correlated with the total parenchyma fraction and the cell and lumen diameters of fibers. Studies suggest that the axial parenchyma acts as a secondary support system for water conduction [34, 35, 38, 63, 76]. The hydraulic vulnerability that vessels acquire with the increase in diameter is compensated by the hydraulic support system formed by the large fraction of parenchyma that protects them against the negative effects of cavitation [34, 62, 63, 69, 70, 76, 78].

Fiber cell and lumen diameter were positively correlated with vessel diameter and vulnerability index. Fiber diameter and lumen are traits that describe the relationship between cell size and the degree of lignification of cell walls and therefore the mechanical-hydraulic resistance characteristics of the wood [32, 33, 69, 77]. Thus, wider fibers with wide lumina have low cell wall areas. Functionally, this kind of fiber provides less mechanical-hydraulic resistance to wood since most of the cellular space is occupied by the empty lumen or by reserve substances, e.g., starch in *Spondias purpurea* [32, 33, 63, 69, 75]. Several studies agree in establishing that it is the proportion assigned to fiber cell walls that is correlated with greater mechanical-hydraulic resistance, wood density and lifespan and not necessarily the fraction assigned to fibers overall [32, 33, 62, 63, 69, 71]. Thus, wide fibers with wide lumina characterize obligate deciduous species that escape drought, which have wide vessels and are characterized by low tolerance to negative pressures on their xylem [19]. In contrast, our data show that the decrease in fiber diameter and the reduction in cell lumen due to thicker walls are found in facultatively deciduous species that may tolerate much stronger negative pressures, as other authors mentioned [62, 63, 71].

Storage and mechanical resistance functions seem to have a strong trade-off since fiber and parenchyma fractions were strongly negatively correlated, as detected in other studies [19, 70, 75]. As [19] mentioned, in wood, the space assigned to the vessels is limited (rarely exceeding 20%), so competition for the morpho-functional space occurs between the fibers and parenchyma, although the fraction of fibers usually dominates [35]. Storage characterizes woods with a rapid resource acquisition profile but usually with low mechanical and cavitation resistance since large fiber fractions are characteristic of robust hydraulic systems that are highly resistant to cavitation [19, 70, 75].

It has been assumed that wood density has close relationships with xylem anatomical traits and that its variation may be related to structural modifications at the cellular level and therefore to the functional performance of wood [34, 75], but this should be supported by future research. In this study, wood density was significantly correlated with all anatomical traits, and these relationships support the differentiation between functional profiles that other studies have characterized [73, 74, 76, 77]. Our results suggest that variation in the total parenchyma fraction and the fiber fraction and fiber traits (size and wall thickness) are the major determinants of wood density. However, our data agree with those of other studies in which it is

established that variation in vessel fraction has little to no effect on wood density given the low fraction that vessels occupy in wood [34, 58, 69].

Studies with a large dataset, covering most lineages of angiosperms, have demonstrated that vessel diameter is driven by plant height, while climatic variables have minimal effects [24–26]. As a tree gets taller, water must be transported along longer pathways, and thus, as Hagen-Poiseuille's law states, resistance to flow increases linearly with increasing conduit length. The increase in vessel diameter from the base to tip counteracts the effect of the generated resistance [26]. With this simple hydraulic principle, it is expected that in a random sample, vessel diameter describes an allometric relationship with tree height. Our results showed a weak correlation between tree height and vessel diameter. This may be for several reasons. First, our sampling has a limited range of heights. Most of our individuals were between 7 and 15 meters tall, so it is possible that the effect of the increase in the pathway was not sufficient to describe this allometric relationship. A recent analysis of the residual variation in vessel diameter using tree height as the main predictor in tropical dry forest trees suggested that tropical dry forest species contain a high variation in vessel diameter residuals and that it is likely that in these seasonal systems, water, as the main limiting factor, has an important effect on vessel diameter [91]. In addition, given the strong selective pressure of water, different functional strategies have arisen in tropical dry forests that allow trees to deal with water deficits. The authors suggest that leaf and wood traits could have a marked effect on vessel diameter independent of tree height. Our results agree with this hypothesis since they suggest that the functional profile is the major determinant of the variation in vessel diameter. For the species studied here, variations in fiber and parenchyma traits were strongly correlated with vessel diameter and were highly consistent across the economic spectrum.

## Conclusions

Our study demonstrates that wood anatomical traits characterize functional groups across the economic spectrum of tropical dry forests and that differences occur in all three cell types that make up wood. Differences across the rainfall gradient within the same vegetation type were low, but the greatest variation occurred in vessel attributes, and low variation was found for fibers and parenchyma. In the tropical dry forests of this study, the functional trade-offs between wood cell types reflect the differentiation between functional groups, that is, between wood density, phenology and water use strategies previously described.

## Supporting information

**S1 Table. Localization and general climate characteristics of the study sites.**
(PDF)

**S2 Table. Traits measured in the wood of six tree species from tropical dry forest.**
(PDF)

**S3 Table. Model selection through the AIC and the LRT.**
(PDF)

**S4 Table. Fixed effect coefficients predicted by the linear mixed model.**
(PDF)

**S5 Table. Multiple pair-wise comparison in EMM from LMM.**
(PDF)

**S6 Table. F values from the ANOVA of each wood trait.**
(PDF)

**S7 Table. Pearson correlation.**
(PDF)

## Acknowledgments

This study is part of the requirements for MVA to obtain a Doctorate of Science degree from the Posgrado en Ciencias Biológicas (PCB), Universidad Nacional Autónoma de México. Thanks to Rodrigo Arriaga Gómez and Fernando Gavito Pérez, for the collection permits and all the support they offered us to sample in Sierra de Montenegro and Sierra de Manantlán, respectively. Thanks to Oscar Rangel and Oscar Sánchez for their support and accompaniment during the field trips in Sierra de Manantlán and to Alicia Rojas-Leal for her support in laboratory work. Comments of two anonymous reviewers are highly appreciated.

## Author Contributions

**Conceptualization:** Marco V. Alvarado, Teresa Terrazas.

**Data curation:** Marco V. Alvarado, Teresa Terrazas.

**Formal analysis:** Marco V. Alvarado.

**Funding acquisition:** Teresa Terrazas.

**Investigation:** Marco V. Alvarado, Teresa Terrazas.

**Methodology:** Marco V. Alvarado, Teresa Terrazas.

**Project administration:** Teresa Terrazas.

**Resources:** Teresa Terrazas.

**Software:** Teresa Terrazas.

**Supervision:** Teresa Terrazas.

**Validation:** Teresa Terrazas.

**Writing – original draft:** Marco V. Alvarado, Teresa Terrazas.

**Writing – review & editing:** Marco V. Alvarado, Teresa Terrazas.

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
