## [Decision Letter · Decision Letter 0]

26 Jun 2023

PONE-D-23-14608Wood anatomical traits differ along the economic spectrum and between sites in the tropical dry forest of MexicoPLOS ONE

Dear Dr. Terrazas,

Thank you for submitting your manuscript to PLOS ONE. After careful consideration, we feel that it has merit but does not fully meet PLOS ONE’s publication criteria as it currently stands. Therefore, we invite you to submit a revised version of the manuscript that addresses the points raised during the review process.

Please carefully consider and respond to the concerns of the reviewers. Please also make all raw data available either in the Appendix or in a public repository.

We look forward to receiving your revised manuscript.

Kind regards,

Yang Yang

Academic Editor

PLOS ONE

“This study is part of the requirements for MVA to obtain a Doctorate of Science degree from the Posgrado en Ciencias Biológicas (PCB), Universidad Nacional Autónoma de México. Thanks to Rodrigo Arriaga Gómez and Fernando Gavito Pérez, for the collection permits and all the support they offered us to sample in Sierra de Montenegro and Sierra de Manantlán, respectively. Thanks to Oscar Rangel and Oscar Sánchez for their support and accompaniment during the field trips in Sierra de Manantlán and to Alicia Rojas Leal for her support in laboratory work. To the Consejo Nacional de Ciencia y Tecnología for the scholarship (810216) awarded to MVA to perform doctorate studies.”

“TT funded by PAPIIT-UNAM in212622, MVA funded by CONACYT 810216

PAPIIT-UNAM, Programa de Apoyo a Proyectos de Investigación e Innovación Tecnológica, Universidad Nacional Autónoma de México.

CONACYT, Consejo Nacional de Ciencia y Tecnología,

https://dgapa.unam.mx/index.php/impulso-a-la-investigacion/papiit

https://conacyt.mx/

6. We note that Figure 1 (A) in your submission contain [map/satellite] images which may be copyrighted. All PLOS content is published under the Creative Commons Attribution License (CC BY 4.0), which means that the manuscript, images, and Supporting Information files will be freely available online, and any third party is permitted to access, download, copy, distribute, and use these materials in any way, even commercially, with proper attribution. For these reasons, we cannot publish previously copyrighted maps or satellite images created using proprietary data, such as Google software (Google Maps, Street View, and Earth). For more information, see our copyright guidelines: http://journals.plos.org/plosone/s/licenses-and-copyright.

a. You may seek permission from the original copyright holder of Figure 1 (A) to publish the content specifically under the CC BY 4.0 license. 

7. We note that Figures 1(B) and 1(C) in your submission contain copyrighted images. All PLOS content is published under the Creative Commons Attribution License (CC BY 4.0), which means that the manuscript, images, and Supporting Information files will be freely available online, and any third party is permitted to access, download, copy, distribute, and use these materials in any way, even commercially, with proper attribution. For more information, see our copyright guidelines: http://journals.plos.org/plosone/s/licenses-and-copyright.

a. You may seek permission from the original copyright holder of Figures 1(B) and 1(C) to publish the content specifically under the CC BY 4.0 license.

b.If you are unable to obtain permission from the original copyright holder to publish these figures under the CC BY 4.0 license or if the copyright holder’s requirements are incompatible with the CC BY 4.0 license, please either i) remove the figure or ii) supply a replacement figure that complies with the CC BY 4.0 license. Please check copyright information on all replacement figures and update the figure caption with source information. If applicable, please specify in the figure caption text when a figure is similar but not identical to the original image and is therefore for illustrative purposes only.

Additional Editor Comments:

Please carefully respond to the concerns of the referees.

Reviewers' comments:

Reviewer's Responses to Questions

**Comments to the Author**

1. Is the manuscript technically sound, and do the data support the conclusions?

Reviewer #1: Partly

Reviewer #2: Partly

2. Has the statistical analysis been performed appropriately and rigorously? 

Reviewer #1: Yes

Reviewer #2: No

3. Have the authors made all data underlying the findings in their manuscript fully available?

Reviewer #1: No

Reviewer #2: Yes

4. Is the manuscript presented in an intelligible fashion and written in standard English?

Reviewer #1: Yes

Reviewer #2: No

5. Review Comments to the Author

Reviewer #1: Summary

The authors measured woody anatomical traits from 6 species at three sites to examine tradeoffs in wood anatomy and how those may relate to function. The work is generally well presented but I had numerous small comments and concerns mostly around clarity, additional detail, and framing their discussion.

Major comments

-In many cases you make very strong statements about how wood anatomy drives function when you do not have any data on function for your species. You can still discuss the potential links between the two but need to be more careful about discussing your data and the potential links your data have to function based on other published studies.

-I am not sure about your modeling approach with AIC. I think it may be a clarification issue, but as I note below in my minor comments, I don’t quite follow the rationale for the approach you took.

Minor comments

• Needs to be reviewed for grammar and spelling.

• Line 66: please note that this is not taking into account vessel length

• Line 68-69: this is still quite controversial. Please see recent discussion in the literature such as: Isasa, Emilie, et al. "Addressing controversies in the xylem embolism resistance–vessel diameter relationship." New Phytologist (2023). At the very least, you should soften this phrasing to say things like “tend to” or “are often associated with”. Also, clarify that it seems to be species with small vessels that tend to have more resistant xylem (rather than smaller vessels are more resistant).

• Most figures and images are of low resolution. Make sure they are high quality for publication.

• Fig 1. Inset graphs too small to read, missing axis labels. Map needs basic requirements of scale bar, north arrow. More detail in caption (ex. red dots = sites).

• Table 1: what is the source of this data? Which years were included?

• Table 2. Source of wood density data? What about variation?

• Line 142: healthy is a very subjective term. Can you provide more exact criteria? Also, do you have any other information on the sample trees like DBH, height, age, etc. Any indication that variation there matters?

• Line 143: what is a “wood slide”?

• Line 144: so you analyze only the last three years of growth? When did you collect the samples?

• Line 151: sounds like dry weight per FRESH volume, right? Please add that clarification.

• Line 164-169. Not enough detail here to replicate this. How did you select the 50 cells to measure? Started with the outermost cell and just worked your way in from there? How did you select a cell to start with? Also, where were these “fields” placed and how was that decided? What does “in” mean? The whole vessel was inside the field or just part of its cell wall would count?

• Table 3 needs some more detail. For example, VF is in relation to all other cells including rays etc.?

• Line 175: need at least one more sentence that briefly explains this method so readers don’t have to read that paper to get the sense for what you did.

• Line 177: per individual, right? So, using your minimum threshold that suggests you coded at least 81,000 points. 900 * 6 species * 5 individuals * 3 sites. Is that correct? And this was all done manually? If so, was there any checking done by someone else to be sure interpretations of fibers vs parenchyma vs vessels were correct?

• Do you know there are no tracheids in these species? If not, you need to explain how those may impact these codes.

• Figure 2C is helpful but a little messy with overlapping circles, gaps, duplicates, etc. Why not also code each point in panel 2B? Wouldn’t that better help us understand? Oh, just realizing 1A has codes and colors in it. They are impossible to see at this scale and make the whole thing appear messy. Why not just do one nice clean image showing the grid points and the cell assignments?

• Line 184-198: glad to see that you accounted for subsampling with your linear mixed model.

• Line 191. Delta AIC of 2 is pretty small. Often we see 4, 8, or even 10 as the thresholds. Why 2? Also, why using AIC? In this case, you have nested models, right? So you can use likelihood ratio test to compare your three models. In fact, with such a simple model formulation, you can just start with a full model and slowly reduce it down. It seems to me your full model should be a species * site interaction. If the interaction is not significant, then go to species + site. Then, if one of those is not significant, you can remove it or just leave it – it shouldn’t matter with your balanced dataset and you can just report the model summary. If you insist on sticking with AIC, please discuss when there are only small differences in AIC and how that impacts the interpretation of results.

• Line 192: normally you only do pairwise of all combinations (species X site in your example) if you test for and find a significant interaction. I see no indication that you considered interactions.

• Line 201: give us a brief summary what this calculation does and how you calculated it (pooled for a species across sites?). If you did pool, how do we know it is not some form of genotypic variability instead of phenotypic? Phenotypic would really be within a site, right? And even then, they may all be expressing similar phenotypes since they are all experiencing similar environmental conditions.

• Line 208: table S2 isn’t easy to read. If you stick with AIC, just show the delta AIC here.

• Fig 3. Expand caption to explain all axes more clearly, remind reader of sample sizes, etc. Seems like you are forcing these groups. Why not just use a continuous axis like wood density across species and use regression?

• Fig 3 stats: I understand your approach but because you are not providing a ton of information about each model, it is hard to assess if they were run correctly. The statistics do not align with the visual error bars in these graphs that would suggest fewer differences than you found. Supposedly this is driven by your log scaling and your random effects for individual within site, right? Can you show us more of those data in a way that clarifies this? Perhaps some discussion around how much variation was accounted for by random effects at each level would help?.

• Line 212: they actually are differences among species

• Figure 5. More detail in caption. What is each dot representing? A sample? An individual? Remind readers so they interpret correctly. Why forcing groups when it appears there are differences by species within a group and the groups overlap so much? Define all abbreviations. Letters for sig differences should always start with a at the top or left side of the graph. The labels on panel A axes are a bit subjective. I suggest you remove terms like “storage capacity” and instead just label it what it is: “parenchyma fraction”. In your discussion you can discuss if you think parenchyma fraction is an estimate of storage capacity.

• Line 261. 30%?

• Line 266 “among”

• Line 277-279. What trait? I don’t really follow this sentence. How are you assessing “fit” here?

• Figure 7 caption. Remind reader what a dot represents. Looks like more dots than individuals here. Is each dot a vessel? If so, how does that impact how you ran your statistics? It actually changes for each row of panels, right? Clarify. Why not a continuous axis like total precip here? Briefly clarify vulnerability index. Why does the order of bars change in each panel? Why does the species change in the middle column? Where are the other species? Overall, I found this figure to be confusing.

• Lines 305-314: So you are only showing the significant ones? This is confusing to me. If showing the non-significant ones takes up too much space, you at least need a clear summary in the main text showing which species had these differences and which did not for all variables tested.

• Figure 8. Similar comments to all other figures about detail in caption. Order of bars changes.

• The results text could be shortened by about 1/3 or more without losing meaning. There is quite a bit of redundancy that is confusing.

• Line 328: Correlations between VI and VF as well as VI and VD are of course going to be strong because VD and VF are used to calculate VI.

• Figure 9. I guess this is okay as it is but be aware that these multipanel figures are asking a lot of readers when the x- and y-axes are shifting a lot and do not align across panels.

• Surprised to see no discussion of intra-species relationships. They clearly differ for many of these interspecies relationships.

• Line 340: confusing – is it weak or not significant?

• Table 4: see my concerns about this index in the methods section.

• Figure 10. Personally, I think that a mixing of figures 10 with figures 4 and/or 6 would be more compelling than this version that is separated. It is okay as it currently is but only the most devoted readers will dive into the details of figures 4 and 6 and I think this figure (10) misses an opportunity to visually depict your species with actual images.

• 393-394: as opposed to lignified parenchyma?

• 394: confusing phrasing around “assigns”

• Line 408-409. Careful with your discussion of hydraulic resistance here (and elsewhere) you did not measure it. So you need to soften your phrasing and/or rely on citations of studies that did measure it.

• Line 513, which studies?

• Line 516: “potentially offering…”

• The discussion is stretching beyond what was measured quite frequently and assuming causation when there is often only correlation or in some cases, the authors don’t even have data. This can be remedied mostly with careful phrasing.

• Line 536: and since there is such a strong desire to draw conclusions about function purely from structure measurements, shouldn’t there also be a recommendation here (or elsewhere) to further our understanding of structure-function relationships for diverse tree species?

• Line 540-546: these are both long complex sentences that are difficult to follow

• 582-583: but you don’t have data on water potential, so can you really say this?

• Line 597: this statement is too broad, subjective, and over-generalized.

• Line 598-600. I don’t think “assumed” is the right approach to this sentence. There is quite a bit of data on how anatomy and density are related and how those drive function

• Line 603: “great confidence” is subjective. Stick to your data and compare it to other published data. You don’t have any data on functional performance so instead rely on other studies that do without bringing in subjective terms like “great confidence” unless you are backing them up with data.

• Line 604: cite these “other studies”

• Line 604 -605: as written, I don’t think your data support this statement and you don’t have it backed up by a citation right now.

• Line 607: is “no effect” correct or is it “little to no effect”?

• Line 608: again, the term “assigned” seems to imply the tree made a decision about what type of cell to make.

• Line 610: stretching too far and too strongly to function. I know what you are trying to argue here but you just don’t have the data to back this up and are speculating about function. You need to soften the phrasing.

• Supporting information needs more detailed captions and/or footnotes so that readers can interpret all abbreviations and some of the context for each table.

• I didn’t see any mention of raw data being posted to a repository for public access. As the authors know, these data are valuable and time-intensive to collect. I would encourage the authors to consider posting the data associated with this publication.

Reviewer #2: COMMENTS FOR THE AUTHOR

GENERAL COMMENTS:

The manuscript by Marco et al. examines variation in anatomical functional traits of wood across 6 tropical species along three different rainfall regimes. In this study, the author has conducted a lot of work in the field and laboratory experiments, and compared trait variation across different environmental conditions by using almost the same tree species (5 out of 6 species). Overall, the manuscript is worthy to be published unless substantial and proper revision. But there remains several drawbacks and need to be addressed very carefully. The research questions are clear but the whole manuscript is “heavy” and wordy, data interpretation is not clear and didn’t follow by an very well structured and concise discussion. The author has stated that the tree height has little effect on vessel diameter in this study, but lack data supports. Furthermore, the main hypotheses and predictions in the manuscript are too general or already well-known in the ecological literature (e.g. efficiency vs safety), so the novel contribution of the study is not very clear, need to be further explained.

SPECIFIC COMMENTS:

TITLE:

“Wood anatomical traits differ along the economic spectrum” sounds confusion, “Wood anatomical traits” should belong to “plant economic spectrum”.

ABSTRACT:

- line 21,“ cell traits” means “anatomical traits”? if so, use term “anatomical traits” is better. Same as the remaining manuscript.

- line 30-32, what do you mean?

- line 35, typo, “study” not “sturdy”

INTRODUCTION:

In the introduction section, you mainly emphasize the vessel diameter, but didn’t explain clearly why you want to/why is it necessary to quantify the anatomical traits, why you aim to compare different rainfall regimes, is it important? Why it’s important to examine the trait variation and trade-offs, what’s the ecological significance or what we can understand based on such trait associations?

-Line 57, what do you mean “end of functional spectrum”, can you explain?

-Line 71-72, further explain the weak relationship between xylem safety and efficiency, why the recent research differ from other studies, these studies are for tropical forest? Which species are they focus? Need to be further explained and compared.

-Line 74, what are the specific traits and references, not just mention “other anatomical traits are …”.

-“trait”--- “traits”

-Line 91-92, “cover the full economic spectrum” sounds too absolute, what do you mean “full”? May use “wide economic spectrum” instead.

METHODS:

- Table1, line 105, more caption information should be added.

Check “coefficient of variation is in parentheses”, 1) for max temperature, there is month information in the parentheses; 2) “coefficient of variation” is calculated based on 12 months of a year? 3) do you mean “coefficient of variation” or “standard deviation”. The coefficient of variation (CV) is defined as the ratio of the standard deviation to the mean, and the unit is not “mm”.

- Figure 1, figure resolution is very low, it’s hard to read the text of subplot.

- Line 115-116, how about “June”, rainy season? Now it’ missing.

- line 125, no unit for 40.9

- line 131, how do you define “extreme of the economic spectrum” and “intermediate species”, do you compare the traits of multiple tree species in all study sites? More information should be provided earlier, e.g., in INTRODUCTION or somewhere in this METHOD section.

- Table 2. Wood density data needs to be aligned.

- line 129, you selected two species for each site, but how many replicates do you have? I.e., how many individuals for each tree species. Such information should be provided.

- line 145, one sample was used to calculate...

- line 142-151, description of wood density measurement can be shorter and concise.

- line 155, use specific alcohol level rather on only “from 50% to 100%”.

- line 156-157, what’s the magnification of the lens.

- line 157, given the important relationship between tree height and vessel diameter, the information of “tree height” should provided somewhere in the Method section, e.g., provide in Table 2.

- line 162, not clear, “80 individuals” mean all tree individuals? Same as line 165, the “individual” shown hear means each thin section for anatomical trait measurement?

- line 163-164, sounds confusion, “their relationships with vessel traits and wood density between functional groups”, why only vessel traits but no fibre wall thickness?

- line 167, confusion, why “10 fields”? You didn’t mention it in earlier text.

- line 168-169, add reference or link for this analyzer.

- line 172, better to use “six tree species”

- Table 3, why you choose these 11 traits, what are their main function? Any supporting references? You may provide such information as text or provide in Table 3.

- line 176, “each individual” means each thin section?

- line 178, add reference or link.

- Fig. 2, nice visualization but the caption is not complete. What do the green dots marked with no.4 refer to?

- line 184, two...”models” not “model”

- Line 191, model with lowerΔAIC is more plausible, do you mean ΔAIC<2.0 rather than >2.0?

- line 192, “were” not “was”

- line 196-197, ANOVA assumes that the residuals from the ANOVA model follow a normal distribution. When you conduct log-transformation, whether the residuals meet or close to the assumption? Basic plots should be provided in the supplementary (e.g., Q-Q plot, histogram of model residuals) to show that transformed data is better.

- Statistics in Method section not clear.

RESULTS:

- line 208, ΔAIC > 2.0? “Table in S2, ΔAIC > 2.0 compared to the null model and p<0.05 in pair-wise comparisons” can be simplified into “Table S2, P<0.05”.

- line 215, “in the study species”--- “of the studied tree species”

- Fig. 4, nice illustration!

- line 230, what do you mean “supported that the null model”, where is your output of LMMs? I failed to find the corresponding outputs in the main manuscript.

- I may delete the line 229-231, and start with “The species of group I had the...”.

- line 231, “The species of group..., even differing between the two species” isn’t logically correct, “trait value” should be the subject, e.g., anatomical trait values differed between the two tree species.

- for the result section, please provide the mean±SD of the trait value, not use “above 200 µm”, “just above 150 µm”, “above 90”.

- line 252, very confused by the sentence “but not between the species of group II and II.”

- line 253-256, what is the difference between the two sentence!! the first sentence refer to a specific species? Why you use “In contrast”?

“Fiber fraction was always close to 70% in group III, while the parenchyma fraction was rarely greater than 30%. In contrast, the parenchyma fraction of Cochlospermum vitifolium was always close to 70% and fiber fraction was rarely greater than 30%.”

- line 267, “Lower case letters in B and C indicate different among species.” not correct. Do you want to say “Different lower case letters in B and C indicate significant differences”?

- line 227, suddenly occurs the “moisture gradient” without any description in Introduction section, why not just use “rainfall gradient” or explain the relationship between moisture and rainfall regimes earlier in Introduction or Method section.

- line 277-279, why? Which figure to support your statement?

- line 303, Fig. 7, not “Lower case letters indicate differences” but “different lower case letters indicate differences...”

- line 305-306, please be aware that you are comparing site variation. “we found statiscally (typo, statistically) significant differences only for L. divaricatum” is not correct, should mention “significant difference among three sites”.

- line 307, typo “statistically” not “statically”

- line 317, “site” not “Site”, add “different lower case letters indicate differences...” for Fig. 8

- line 324, if you say that there is a negative correlation, please add (coefficient ; P-value)

- Fig. 9, it’s obvious that the correlation is mainly driven by groups, how about the trait association within each group, will it differ a lot? Why?

- Fig. 10, blue color indicates parenchyma cells while grey color indicates fibre cells? Where is the vessel information shown in figure 10? current version is not complete, figure 10 needs to be further adjusted for better visualization.

- it’s a “heavy” result section, too many figures and tables, maybe remove part of them to supporting material, e.g., Table 1. General site information & Table 3 basic trait information, and some figures.

DISCUSSION:

- line 395, “wood volume” rather than “wood”.

- line 420, remove “very” or change to “very thick walls”

- line 448-449, what do you mean “group II fibers are characterized by their very thick walls allowing them to vary parenchyma fraction.” group III also has thickened fibres but why group III has low parenchyma fraction? Is this just your personal defination or there is any causal relationship between thickened fibres and parenchyma?

Line 452, avoid redundancy, delete “As mentioned in the results, for”

- line 462-263, why “since”, I didn’t find the causal relationship between “small changes in vessel diameter have important effects” and “the largest variation in vessel diameter and frequency”

-line 465-466, mention which three species.

Line 497, change “what was previously thought” to “expected”

-line 511, which two species? Need to be mentioned

- line 513, have you already read all related studies and found the same conclusion that significant variation between fibre and parenchyma? Be careful to use the term “all studies”, which sounds too extreme, maybe you can use “previous studies have shown that [with references]”

- Line 533-534, I didn’ get it, can you further explain “ there is low variation in these functions and therefore the greatest selection operates on the vessel traits.”

- line 635, should mention which three cell types

- it’s good to see that the author discussed the relationship between tree height and vessel diameter and mentioned that there is a weak correlation between tree height and vessel diameter in this study, however, no data was shown to support the statement, it’s necessary to show the height information somewhere in the manuscript and conduct basic statistics to test the effect of tree height on vessel diameter.

6. PLOS authors have the option to publish the peer review history of their article (what does this mean?). If published, this will include your full peer review and any attached files.

Reviewer #1: No

Reviewer #2: No

---

## [Author Response · Author response to Decision Letter 0]

23 Aug 2023

Reviewers 

We appreciate your comments and here are only the general ones. The specific comments for the manuscript are given in the response to reviewers letter 

Reviewer #1: Summary

The authors measured woody anatomical traits from 6 species at three sites to examine tradeoffs in wood anatomy and how those may relate to function. The work is generally well presented but I had numerous small comments and concerns mostly around clarity, additional detail, and framing their discussion.

Thanks for your positive comments.

Major comments

-In many cases you make very strong statements about how wood anatomy drives function when you do not have any data on function for your species. You can still discuss the potential links between the two but need to be more careful about discussing your data and the potential links your data have to function based on other published studies.

We appreciate all of your comments that you have made to improve this manuscript. We have read each of your comments and carefully review the text to include all pertinent corrections. We are aware that our discussion is heavy and wordy. We have carefully revised it to reduce its length. We are also aware that there is no direct equivalence between wood anatomical traits and function. Even so, we have been more careful in our statements and we re-write several of them in the discussion. However, all our discussion is supported by other works that, like us, study the same anatomical attributes and reach similar conclusions [e.g. 31, 34, 35, 38, 51, 62, 69, 70, 75, 76, 77]; further, although there are fewer of them, there are studies that combine analysis of wood anatomical traits and physiological measurements and we support our discussion on them [e.g. 18, 19, 22, 33, 34, 57, 61, 63]. 

-I am not sure about your modeling approach with AIC. I think it may be a clarification issue, but as I note below in my minor comments, I don’t quite follow the rationale for the approach you took.

We understand your concerns regarding our statistical approach. We have responded to each of your minor comments regarding this and listened to your suggestions and observations. However, we want to make a couple of clarifications in this comment.

1) We do include the interaction of both factors (site and species) as part of our models, we regret this mistake. Now, in table S2 we already include the output of this model. We also carried out a model with only the site as a factor, we mentioned it in our results but we had not included it in table S2 for the simplicity of the table. Now it is also present.

2) although ΔAIC and the likelihood ratio test (LTR) have different approaches, for our purposes either can be used. ΔAIC indicates whether a more complex model is more plausible, while LTR penalizes more complex models that do not explain more variation than they predict. Either way, using AIC or LTR for model selection, both approaches support the same models (see table in S2). The model with the interaction is always the best supported, followed by the model with only the effect of the species. The only difference is that in the model in which the effect of both factors is added, it is penalized by LTR and is not statistically significant. This can be explained because the effect that the site adds by itself explains little variation by itself, so for the TLR adding this effect without interaction penalizes the model and is not significant.

3) Although in our methods we consider a ΔAIC of 2, the value we obtained from the AIC in the comparison of models is greater than 40. So the differences between models are not small.

Reviewer 2

GENERAL COMMENTS:

The manuscript by Marco et al. examines variation in anatomical functional traits of wood across 6 tropical species along three different rainfall regimes. In this study, the author has conducted a lot of work in the field and laboratory experiments, and compared trait variation across different environmental conditions by using almost the same tree species (5 out of 6 species). Overall, the manuscript is worthy to be published unless substantial and proper revision. But there remains several drawbacks and need to be addressed very carefully. The research questions are clear but the whole manuscript is “heavy” and wordy, data interpretation is not clear and didn’t follow by an very well structured and concise discussion. The author has stated that the tree height has little effect on vessel diameter in this study, but lack data supports. Furthermore, the main hypotheses and predictions in the manuscript are too general or already well-known in the ecological literature (e.g. efficiency vs safety), so the novel contribution of the study is not very clear, need to be further explained.

Thanks for your comments. Now, we expect that the revised manuscript resolved the drawbacks since it was rewritten along the text and it has been edited carefully.

---

## [Decision Letter · Decision Letter 1]

28 Sep 2023

PONE-D-23-14608R1Wood anatomical traits differ along the plant economic spectrum and between sites in the tropical dry forest of MexicoPLOS ONE

Dear Dr. Terrazas,

Thank you for submitting your manuscript to PLOS ONE. I appreciate your time on revising the manuscript, and apologize for the relatively long time of the review process. The reviewer has pointed out a few places for more attention. So I am returning this manuscript as minor revision. 

We look forward to receiving your revised manuscript.

Kind regards,

Yang Yang

Academic Editor

PLOS ONE

Journal Requirements:

Reviewers' comments:

Reviewer's Responses to Questions

**Comments to the Author**

1. If the authors have adequately addressed your comments raised in a previous round of review and you feel that this manuscript is now acceptable for publication, you may indicate that here to bypass the “Comments to the Author” section, enter your conflict of interest statement in the “Confidential to Editor” section, and submit your "Accept" recommendation.

Reviewer #2: All comments have been addressed

2. Is the manuscript technically sound, and do the data support the conclusions?

Reviewer #2: Yes

3. Has the statistical analysis been performed appropriately and rigorously? 

Reviewer #2: Yes

4. Have the authors made all data underlying the findings in their manuscript fully available?

Reviewer #2: Yes

5. Is the manuscript presented in an intelligible fashion and written in standard English?

Reviewer #2: Yes

6. Review Comments to the Author

Reviewer #2: COMMENTS FOR THE AUTHOR

GENERAL COMMENTS:

Marco et al. Has addressed most of my comments and largely improved the manuscript. But here are some minor questions still need to be solved.

SPECIFIC COMMENTS:

Lines you mentioned in the response letter don’t match the updated manuscript, need to check carefully.

-Line 71-72, further explain the weak relationship between xylem safety and efficiency,

why the recent research differ from other studies, these studies are for tropical forest?

Which species are they focus? Need to be further explained and compared.

(Author replied before: We modify the sentences to clarify the statement. Now lines 78-82.)

Q1. However, no changes were made between the original one and revised ms

-Line 74, what are the specific traits and references, not just mention “other anatomical

traits are …”-“trait”--- “traits”

(Author replied before: We now mentioned which ones in general, lines 84-85. )

Q2. could you check it again, why you didn’t make any change. Lines 84-85 (updated MS) are totally irrelevance of what I asked about Line 74 (original MS). I can still see that line 76 in your updated manuscript you just mention “other anatomical traits”, which is unclear and what specific traits you refer to.

- Fig. 9, it’s obvious that the correlation is mainly driven by groups, how about the trait

association within each group, will it differ a lot? Why?

(Author replied before: As mentioned for reviewer 1, based on the main aims of this study we consider that correlations within the species will not contribute to our discussion. )

Q3. I am afraid that I don’t think so. I can understand that you want to you look at all studied tree species, but you have already divided different tree species into three groups in the manuscript, and colored them into different colors in the scatter plots. For a reader, I will be curious how these relationship will change if look at each group. So I suggest:

Suggestion 1: if you are not interested to look at each group, then use the same dot color for all tree species. Or

Suggestion 2: provide correlations within each group and correlation lines with corresponding colors, while use black line to show the general correlation as you showed before. Or

Suggestion 3: provide scatter plots to show the traits association within each group (Group I, Group II, Group III) in the supplementary.

Q4. please check the order Figure 3 and Figure 4 (they exchanged compared to the original manuscript).

Q5. The title of the ms still need to be modified.

“Wood anatomical traits differ along the plant economic spectrum” is not right.

Here is one suggestion:

“Tree species differ in plant economic spectrum traits in tropical dry forest of Mexico”

7. PLOS authors have the option to publish the peer review history of their article (what does this mean?). If published, this will include your full peer review and any attached files.

Reviewer #2: No

---

## [Author Response · Author response to Decision Letter 1]

10 Oct 2023

GENERAL COMMENTS:

Marco et al. Has addressed most of my comments and largely improved the manuscript. But here are some minor questions still need to be solved.

SPECIFIC COMMENTS:

Lines you mentioned in the response letter don’t match the updated manuscript, need to check carefully.

-Line 71-72, further explain the weak relationship between xylem safety and efficiency, why the recent research differ from other studies, these studies are for tropical forest?

Which species are they focus? Need to be further explained and compared.

(Author replied before: We modify the sentences to clarify the statement. Now lines 78-82.)

Q1. However, no changes were made between the original one and revised ms

We appreciate this comment and apologize for this omission. We have delved a little deeper into contrasting the controversy between xylem safety and efficiency.

-Line 74, what are the specific traits and references, not just mention “other anatomical

traits are …”-“trait”--- “traits”

(Author replied before: We now mentioned which ones in general, lines 84-85. )

Q2. could you check it again, why you didn’t make any change. Lines 84-85 (updated MS) are totally irrelevance of what I asked about Line 74 (original MS). I can still see that line 76 in your updated manuscript you just mention “other anatomical traits”, which is unclear and what specific traits you refer to.

In relation to this comment, this final sentence of the paragraph served as a connection and introduction to the following paragraph where we commented on studies in which anatomical traits are analyzed in addition to vessel diameter and frequency. We consider that joining these two paragraphs respond to reviewer 2. Now the whole paragraph allows us to show that we do mention other works in which different anatomical traits are analyzed that are important to understand plant hydraulics. 

- Fig. 9, it’s obvious that the correlation is mainly driven by groups, how about the trait association within each group, will it differ a lot? Why?

(Author replied before: As mentioned for reviewer 1, based on the main aims of this study we consider that correlations within the species will not contribute to our discussion. )

Q3. I am afraid that I don’t think so. I can understand that you want to you look at all studied tree species, but you have already divided different tree species into three groups in the manuscript, and colored them into different colors in the scatter plots. For a reader, I will be curious how these relationship will change if look at each group. So I suggest:

Suggestion 1: if you are not interested to look at each group, then use the same dot color for all tree species. Or

Suggestion 2: provide correlations within each group and correlation lines with corresponding colors, while use black line to show the general correlation as you showed before. Or

Suggestion 3: provide scatter plots to show the traits association within each group (Group I, Group II, Group III) in the supplementary.

We appreciate your concerns regarding this figure and understand your point. We decided to color each point in relation to the functional groups precisely to show how the correlation between wood traits follows the economic spectrum continuum proposed by the groups and that the individuals of the species are not distributed randomly. Color the points allowed us to show not only that there is a strong correlation between traits, but also that this correlation follows the functional spectrum. We understand that doubts may arise as to whether this same pattern holds within the groups. We consider this point will not modify the purpose of this paper; because within each group there are only the individuals of two species, so it is relatively obvious that the correlations will decrease given that the individuals of the species are much more similar in their traits than with respect to the rest of the species. See an example of the graphs for vessel diameter vs vessel frequency within each group. Thus, we consider that the pattern that exists in the economic spectrum of tropical dry forests is approximated with complete data. This is why we have decided to use the same color for all points in order to maintain only this interpretation. The figure was modified and now all the points are in black.

Q4. please check the order Figure 3 and Figure 4 (they exchanged compared to the original manuscript).

We have reviewed the order of the figures. It is true that during the editing of the manuscript we changed the order of these two figures. However, we have been careful to properly cite each one of them and place the appropriate captions on each figure. They are correct.

Q5. The title of the ms still need to be modified.

“Wood anatomical traits differ along the plant economic spectrum” is not right.

Here is one suggestion:

“Tree species differ in plant economic spectrum traits in tropical dry forest of Mexico”

We appreciate the suggestion and modify the title. It is a striking title and retains the essence of the focus

---

## [Editor Report · Decision Letter 2]

12 Oct 2023

Tree species differ in plant economic spectrum traits in the tropical dry forest of Mexico

PONE-D-23-14608R2

Dear Dr. Terrazas,

We’re pleased to inform you that your manuscript has been judged scientifically suitable for publication and will be formally accepted for publication once it meets all outstanding technical requirements.

Kind regards,

Yang Yang

Academic Editor

PLOS ONE

---

## [Editor Report · Acceptance letter]

18 Oct 2023

PONE-D-23-14608R2 

Tree species differ in plant economic spectrum traits in the tropical dry forest of Mexico 

Dear Dr. Terrazas:

I'm pleased to inform you that your manuscript has been deemed suitable for publication in PLOS ONE. Congratulations! Your manuscript is now with our production department. 

Kind regards, 

on behalf of

Dr. Yang Yang 

Academic Editor

PLOS ONE